# Sublethal Effects of Spirotetramat, Cyantraniliprole, and Pymetrozine on *Aphis gossypii* (Hemiptera: Aphididae)

**DOI:** 10.3390/insects15040247

**Published:** 2024-04-03

**Authors:** Se Eun Kim, Hyun Kyung Kim, Gil Hah Kim

**Affiliations:** Department of Plant Medicine, College of Agriculture, Life and Environment Science, Chungbuk National University, Cheongju 28644, Republic of Korea; rhaqhd7436@naver.com (S.E.K.); nshk0917@gmail.com (H.K.K.)

**Keywords:** *Aphis gossypii*, cyantraniliprole, pymetrozine, life table parameter, malformation, spirotetramat, sublethal effect

## Abstract

**Simple Summary:**

This study investigated the sublethal effects of three insecticides (spirotetramat, cyantraniliprole, and pymetrozine) on *Aphis gossypii*. The effects of sublethal concentrations (LC_10_, LC_30_, LC_50_, and LC_70_) of the insecticides on the developmental period, survival rate, adult longevity, fecundity, and deformity rate were compared with those of the control. Spirotetramat and cyantraniliprole caused malformation in the F_1_ but not the F_2_ generation of *A*. *gossypii*. The net reproductive rate (*R*_0_) decreased significantly compared to that of the control (43.8) for all insecticides except cyantraniliprole at the LC_10_ (37.5). Therefore, sublethal concentrations (over the LC_30_) of the three insecticides could aid in the management of *A. gossypii* by affecting its population density.

**Abstract:**

The toxicity and sublethal effects of three insecticides (spirotetramat, cyantraniliprole, and pymetrozine) on *Aphis gossypii*, a major agricultural pest, were investigated. The nymphal stage showed greater susceptibility than the adult stage to all the insecticides, with a difference of up to 8.9 times at the LC_50_ of spirotetramat. The effects of sublethal concentrations (LC_10_, LC_30_, LC_50_, and LC_70_) of the insecticides on the on the developmental period, survival rate, adult longevity, fecundity, and deformity rate were compared with those of the control. Compared with the control, cyantraniliprole and pymetrozine did not significantly affect the developmental period in the parental or F_1_ generation when applied at the nymphal stage at any concentration. Nonviable nymphs occurred in the F_1_ generation when both nymphs and adults were treated with spirotetramat and cyantraniliprole but not in the F_2_ generation. The age-specific maternity (*l_x_m_x_*) of *A. gossypii* treated with sublethal concentrations (LC_10_, LC_30_) decreased with increasing concentration. Spirotetramat at the LC_30_ resulted in significant differences in all life table parameters (*R*_0_, *r_m_*, *λ*, *T*, *DT*) compared with those of the control. Similarly, compared with that of the control (43.8), the net reproductive rate (*R*_0_) significantly decreased for all the insecticides except cyantraniliprole at the LC_10_ (37.5). Therefore, this study indicated that sublethal concentrations (over the LC_30_) of spirotetramat, cyantraniliprole, or pymetrozine might be useful for the density management of *A. gossypii*.

## 1. Introduction

The cotton aphid *Aphis gossypii* is distributed worldwide and is a major agricultural pest that causes damage to many crops [1]. Chemical pesticides are widely used to control aphids, but pesticide resistance is increasing due to their indiscriminate use [2,3,4,5,6]. The effect of pesticides is reduced by abiotic factors, and physiological and behavioral changes such as longevity, fertility, feeding, and oviposition in individuals exposed to low doses are called sublethal effects [7,8,9]. To use pesticides effectively, it is important to evaluate changes at sublethal doses as well as at appropriate doses [10,11]. Exposure of *Bemisia tabaci* to sublethal concentrations of imidacloprid and bifenthrin resulted in significantly lower honeydew excretion and fecundity levels than those in the control [9]. Treatment of the cabbage aphids *Brevicoryne brassicae* and *Frankliniella occidentalis* with spirotetramat at sublethal concentrations prolonged the preadult development duration, while the preadult survival, adult longevity, and reproduction of the F_1_ generation decreased [12,13]. At sublethal concentrations, chlorantraniliprole decreased the fecundity of *Spodoptera frugiperda*, *S*. *cosmioides*, and *Helicoverpa armigera*, which required a prolonged developmental period, resulting in significantly lower population growth parameters than those in the control group [14,15,16]. Treatment of *A. gossypii* with sublethal concentrations of flupyradifurone also prolonged the prereproductive period but significantly decreased fecundity compared to that of the control [17]. Treatment of the wheat aphids *Sitobion avenae* and *Rhopalosiphum padi* with sublethal concentrations of sulfoxaflor did not affect the parent generation, but *S*. *avenae* showed a decrease in adult longevity, while *R. padi* responded by increasing the intrinsic rate of increase (*r*_m_) and the finite rate of increase (*λ*) of the first progeny generation [18]. A sublethal effect can be positive or negative, but it is a useful pest management strategy for suppressing pest population growth.

Spirotetramat is a cyclic keto-enol insecticide that inhibits insect acetyl-CoA carboxylases (ACCs), interrupting lipid biosynthesis [19,20,21]. It has a systemic effect on plants and is an effective insecticide against piercing-sucking insects, such as aphids, mites, and white flies [21,22]. Spirotetramat is an insect growth regulator (IGR) insecticide that inhibits the growth of young insects at the juvenile and immature stages; reduces reproductive ability; and ultimately causes insecticidal activity [21,22,23,24]. Cyantraniliprole is a diamide insecticide that acts as a ryanodine receptor (RyR) modulator and shows greater selectivity for insect than mammalian RyRs [25,26,27]. It is used for the control of chewing and sucking insects, such as whiteflies, thrips, aphids, and fruit flies [28,29,30,31,32]. Pymetrozine is a representative insecticide of the pyridine azomethine class that was developed for the control of pests such as aphids in crops such as cotton and citrus and has high selectivity and minimal negative impact on mammals, birds, and beneficial insects [33]. It is a nutritional deterrent that acts as a feeding inhibitor on sucking pests, preventing them from ingesting nutrients, and is a highly selective insecticide [34,35].

This study investigated the insecticidal activity of three insecticides (spirotetramat, cyantraniliprole, and pymetrozine) and the effects of their sublethal concentrations (LC_10_, LC_30_, LC_50_, and LC_70_) on the growth parameters of the parent and filial generations of *A. gossypii*.

## 2. Materials and Methods

### 2.1. Insect

*A. gossypii* were collected in May 2022 from the Hwasung area, Gyunggi Province, Republic of Korea. They were reared in acrylic cages (30 × 30 × 30 cm) on 2-week-old cucumber plants. The rearing conditions were 23 ± 1 °C, 50 ± 10% relative humidity (RH) and a 16:8 h light: dark (L:D) cycle.

### 2.2. Chemicals

Spirotetramat (99.4%) was obtained from Bayer Crop Science (Leverkusen, Germany), cyantraniliprole (97.59%) from FMC Corporation (Philadelphia, PA, USA), and pymetrozine (98.3%) from Syngenta (Basel, Switzerland).

### 2.3. Insecticidal Toxicity Experiment

The toxicity of the three tested pesticides to *A. gossypii* was investigated at various concentrations (spirotetramat 0.11–110.0 ppm, cyantraniliprole 0.02–50.0 ppm, pymetrozine 0.65–670.0 ppm) to determine the lethal and sublethal concentrations. The experiment was conducted using the leaf dipping method with cucumber leaves. Briefly, leaves (ø 5 cm) were immersed in the diluted pesticide for 30 s and then dried for 2 min. The dried leaves were subsequently placed in a Petri dish (ø 5 cm) and 20 adults or 20 nymphs were then placed on the leaves. First-instar nymphs of *A. gossypii* born within 24 h from adults inoculated on untreated leaves were used in the experiment. Additionally, adults of *A. gossypii* within 12 h of emergence were also used in the experiment. All the experimental insects were incubated at 23 ± 1 °C, 50 ± 10% RH, and a 16:8 h (L:D) photoperiod and mortality was observed for 96 h at 24 h intervals. For the control treatment, the leaves were immersed in distilled water not treated with any pesticide. Mortality was corrected by Abbott’s formula, and all experiments were performed with three replicates.

### 2.4. Development and Reproduction of A. gossypii at Sublethal Concentrations

The effects of the pesticides on the development and reproduction of *A. gossypii* were observed. The experiment was conducted using the leaf dipping method with cucumber leaves. A total of 30 adults and 30 nymphs were inoculated on cucumber leaves immersed in the pesticides at various sublethal concentrations (LC_10_, LC_30_, LC_50_, and LC_70_). Adults and nymphs were exposed to pesticide-treated leaves for 24 h and then individually transferred to Petri dishes (ø 3.5 cm) containing untreated leaves. The developmental period, survival rate, adulthood period, and fertility status of the adults were evaluated for the parental (P) and F_1_ generations. The leaves were replaced every two days. Malformation in the F_1_ and F_2_ generations was investigated after insecticide treatment in nymphs and adults of *A. gossypii*, respectively. All the experiments were performed with 10 replicates and were incubated at 23 ± 1 °C, 50 ± 10% RH, and a 16:8 h (L:D) photoperiod. All experiments were conducted until the last insect died.

### 2.5. Measurement of the Growth Parameters of A. gossypii after Treatment with Sublethal Concentrations of the Three Pesticides

The life table parameters of age-specific survival (*lx*), fecundity (*mx*), and net maternity (*lxmx*), were estimated using sublethal concentrations (LC_10_ and LC_30_).

The fertility life table parameters of net reproductive rate (*R*_0_), intrinsic rate of population increase (*r_m_*), finite rate of population increase (*λ*), generation time (*T*), and doubling time (*DT*), were estimated by Wu et al. [14]. *A. gossypii* were followed from oviposition until death to study longevity and the number of nymphs laid per female in a day.

### 2.6. Data Analysis

The LC values associated with the three pesticides for *A. gossypii* were calculated using probit analysis in SAS [36]. The sublethal effects at sublethal concentrations were arcsine square root-transformed for analysis of variance (ANOVA). Means were compared and analyzed using Tukey’s studentized range test at *p* = 0.05 [36]. The fertility life table parameters at the LC_10_ and LC_30_ (*R*_0_, *T, r_m_*, *λ*, *DT*) were estimated [14]. The data were statistically analyzed using one-way ANOVA followed by Duncan’s multiple range test [36]. The differences in parameters between LC_10_ and LC_30_ of each insecticide were analyzed using a *t*-test in SAS [36].

## 3. Results

### 3.1. Toxicity to A. gossypii at the Recommended Concentrations of the Three Insecticides

The toxicity of the three insecticides to *A. gossypii* nymphs and adults was compared at lethal concentrations (Table 1). All the insecticides had greater effects on the nymphs than on the adults. The difference in susceptibility across developmental stages was 8.0 times at the LC_10_, whereas it was 9.5 times at the LC_90_ for spirotetramat. There was a 2.5-fold difference in the LC_10_ between *A. gossypii* nymphs and adults but a 5.4-fold difference in the LC_90_ for cyantraniliprole. The difference in the lethal concentration of pymetrozine between nymphs and adults of *A. gossypii* was not high, but the LC_90_ showed the greatest difference at 2.3 times. As the concentration increased, the difference in lethal concentration between adults and nymphs also increased for all three insecticides.

### 3.2. Sublethal Response of A. gossypii to the Three Insecticides

The developmental period, survival rate, adult longevity, fecundity, and deformity rate caused by treatment with sublethal concentrations of *A. gossypii* nymphs were studied (Table 2). The developmental period to adulthood did not significantly differ from that of the control at the LC_10_ and LC_30_ values of spirotetramat (F = 14.14; df = 4; *p* < 0.0004), but the survival rate (F = 237.57; df = 4; *p* < 0.0001) and fecundity (F = 280.92; df = 4; *p* < 0.0001) decreased as the concentration increased in parent generation. The deformity rate of the F_1_ generation increased as the insecticide concentration increased (F = 56.24; df = 4; *p* < 0.0001), and the developmental period also significantly differed (F = 23.11; df = 4; *p* < 0.0001). Cyantraniliprole treatment during the nymphal stage did not affect the developmental period until adulthood for either the parents (F = 1.59; df = 4; *p* = 0.2502) or the F_1_ generation (F = 2.28; df = 4; *p* = 0.1325), depending on the treatment concentration. Compared with that of the control, the deformity rate of the F_1_ generation was significantly different above the LC_50_ value (F = 14.84; df = 4; *p* = 0.0003). The adult survival rate (P:F = 20.99; df = 4; *p* < 0.0001; F_1_: F = 42.46; df = 4; *p* < 0.0001) and fecundity (P:F = 227.07; df = 4; *p* < 0.0001; F_1_: F = 44.51; df = 4; *p* < 0.0001) according to pymetrozine concentration differed between the P and F_1_ generations, but no deformities were observed in the F_1_ generation, and the developmental period until adulthood (P:F = 2.51; df = 4; *p* = 0.1083; F_1_: F =1.22; df = 4; *p* = 0.3615) was not significantly different from that of the control. None of the insecticides caused deformities on the F_2_ generation. 

The insecticides were applied during the adult stage of *A. gossypii* and their effects on development were investigated (Table 3). As the spirotetramat treatment concentration increased, the survival rate (P:F = 166.06; df = 4; *p* < 0.0001; F_1_: F = 1048.78; df = 4; *p* < 0.0001), longevity (P:F = 31.46; df = 4; *p* < 0.0001), and fecundity (P:F = 53.54; df = 4; *p* < 0.0001; F_1_: F = 194.96; df = 4; *p* < 0.0001) decreased in both the P and F_1_ generations, but the deformity rate of the F_1_ generation increased (F = 64.56; df = 4; *p* < 0.0001). The survival rate (F = 69.43; df = 4; *p* < 0.0001) and longevity (F = 14.32; df = 4; *p* = 0.0004) of *A. gossypii* adults decreased as the concentration of cyantraniliprole used for the adult stage increased, but there was significant difference in terms of fecundity (P:F = 28.77; df = 4; *p* < 0.0001; F_1_: F = 42.04; df = 4; *p* < 0.0001) compared to that of the control. The developmental period of the F_1_ generation was significantly different from that of the control, but no differences were observed depending on the treatment concentration (F = 4.36; df = 4; *p* = 0.0269). Pymetrozine significantly differed from the control in terms of *A. gossypii* development depending on the treatment concentration (F = 7.68; df = 4; *p* = 0.0043), but no deformities were observed in the F_1_ generation. After treatment of the *A. gossypii* adults with the three insecticides, no deformities were observed in the F_2_ generation.

### 3.3. Effects of Insecticides on the Population Growth of A. gossypii

The *l_x_* and *m_x_* of *A. gossypii* first instars exposed to sublethal concentrations (LC_10_ and LC_30_) of the three pesticides were determined (Figure 1).

Compared with those in the control group, all the insecticide treatments reduced the *l_x_*, *m_x_*, and *l_x_m_x_* values. All the population growth parameters also decreased in the treatment with the LC_30_ compared to those in the treatment with the LC_10_ for the three insecticides. In particular, compared with the LC_10_ treatment, the LC_30_ treatment with spirotetramat led to a greater decrease in the three parameters. The difference in survival times varied depending on the LC_10_ and LC_30_ for each insecticide, with that for spirotetramat being 10 d, for cyantraniliprole being 1 d, and for pymetrozine being 6 d. The highest peaks in *l_x_m_x_* were 4.9 for the control, 1.6 for spirotetramat, 1.6 for cyantraniliprole, and 1.3 for pymetrozine at the LC_30_.

The fertility life table parameters of the F_1_ generation of *A. gossypii* nymphs treated with sublethal concentrations (LC_10_, LC_30_) of the three insecticides are shown in Table 4. *R*_0_ decreased from the LC_10_ to the LC_30_ for both spirotetramat and cyantraniliprole. For spirotetramat, there were significant differences in population growth parameters between the LC_10_ and LC_30_ treatments, except for generation time (*T*) and doubling time (*DT*). For cyantraniliprole, there was a significant difference in the *R*_0_ values (*p* = 0.0405) between the LC_10_ and LC_30_ treatments, but the other parameters showed no differences. For pymetrozine, there was no difference in any population parameters between the two concentrations. The parameters (*R*_0_:F = 16.69; df = 6; *p* < 0.0001; *r_m_*:F = 3.76; df = 6; *p* = 0.0035; *λ*:F = 4.27; df = 6; *p* = 0.0014) were lower than those of the control group for all insecticides except for *T* in the cyantraniliprole LC_10_ treatment (F = 5.12; df = 6; *p* = 0.0003). *DT* did not significantly differ between the treatment and control for any of the insecticides (F = 0.98; df = 6; *p* = 0.4457).

## 4. Discussion

When selecting an insecticide for pest control, its sublethal effects are an important consideration, and outbreaks of *A. gossypii* can be caused by the insecticide eliminating natural enemies or stimulating reproduction [37,38,39,40]. Spirotetramat, cyantraniliprole, and pymetrozine have different insecticidal modes of action but are widely used for *A. gossypii* control because they are insecticides suitable for controlling sucking pests [21,26,41].

In the present study, *A. gossypii* exposed to spirotetramat and cyantraniliprole exhibited malformations in the filial generation. Spirotetramat is a lipid synthesis inhibitor that affects immature stages, reducing fecundity and fertility and thereby reducing insect populations [21]. Spirotetramat is thought to cause deformities by affecting embryonic development and metamorphosis through the inhibition of lipid biosynthesis, and deformities have also been observed in *Myzus persicae* and *Spodoptera littoralis*; in *Frankliniella occidentalis* malformation of the egg structure affected embryonic development and caused the death of eggs before hatching [42,43,44]. In this study, the filial generation of *A. gossypii* treated with spirotetramat were born dead and deformed, with no appendages such as antennae or legs. Lipids, an energy source for developing embryos, are important for overwintering and are essential for growth and reproduction [45,46,47,48]. A decrease in lipids was observed in *S*. *littoralis* larvae treated with chlorfluazuron, in the endoparasitoid *Pimpla turionellae* treated with cypermethrin, in *Hippodamia variegate* larvae treated with spirodiclofen, in the susceptible beetle *Rhyzopertha dominica* treated with deltamethrin, and in *Periplaneta americana* treated with allethrin [49,50,51,52,53]. This indicates that lipid synthesis can affect insecticidal activity.

Cyantraniliprole is a ryanodine receptor insecticide used for managing chewing insects [28,54]. Seed treatment reduced the infestation of the rice water weevil *Lissorhoptrus oryzophilus* and showed high insecticidal activity against *Agrotis ipsilon* larvae when treated with maize plants [55,56,57]. The fecundity, fertility, feeding, oviposition, and mating of *F*. *occidentalis* were affected by cyantraniliprole, which was highly toxic to *H*. *assulta*, as indicated by decreased the percentage of pupating larvae and increased that of deformed adults at sublethal concentrations (LC_5_, LC_15_, and LC_30_) [30,58]. Treatment of *Plutella xylostella* larvae with cyantraniliprole at sublethal concentrations (LC_10_ and LC_25_) increased the occurrence of deformities in adult wings in the parental generation [59]. Cyantraniliprole significantly decreased the *r_m_*, *λ*, and *DT* in the tobacco budworm, *Helicoverpa assulta*, and when the parental generation was treated with the LC_30_, pupal weight and adult fecundity decreased, while adult deformity increased [58]. A significant decrease in the survival rate, longevity, and fecundity of the parental and F_1_ generations compared to those of the control group up to the LC_30_ sublethal concentration was observed in *A. gossypii* adults exposed to spirotetramat and cyantraniliprole. Malformation also occurred above the LC_30_, but more malformations were observed in *A. gossypii* individuals treated with insecticides during the adult stage than during the nymphal stage; therefore, it is thought that the deformities in the F_1_ generation were related to embryonic development.

In addition, the sublethal concentrations of the three insecticides (spirotetramat, cyantraniliprole, and pymetrozine) affected the population growth parameters; in particular, the LC_30_ significantly decreased all the parameters (*R*_0_, *T*, *r_m_*, *λ*, *DT*) compared to those in the control group. These findings showed that the insecticide treatments resulted in a reduction in population density and had a control effect on the next generation. However, the intrinsic rate of increase did not increase or decrease in *A. gossypii* exposed to sublethal concentrations of bifenthrin, acephate, carbofuran, and pyriproxyfen [10]. When exposed to sulfoxaflor at the at the LC_20_, the *λ* and *R*_0_ of the parental generation (G_0_) decreased significantly in *A. gossypii* [60]. The *T* of G_1_ and G_2_ increased, and the reproductive period of G_3_ and G_4_ and the fecundity of G_1_ and G_2_ were greater than those in the control. At the LC_30_ of imidacloprid and pymetrozine for the cabbage aphid *Brevicoryne brassicae*, the net fecundity rate and intrinsic birth rate decreased due to the sublethal effect, and the *r_m_*, *T* and *DT* were lower than those of the control [61]. There was a significant difference in the longevity of females between the control group and the two insecticide groups, but there was no significant difference in life table parameters between the two insecticide groups [61]. The LC values for the toxicity of imidacloprid and pirimicarb to *A. gossypii* increased with age, and both insecticides had negative effects on the several parameters (*r_m_*, *R*_0_, *Ta*, and *λ*) [62]. However, sublethal concentrations also affect natural enemies, and chlorpyrifos treatment of the Asian lady beetle *Harmonia axyridis* significantly increased the preoviposition period, decreasing population growth parameters (*λ*, *r*, *R*_0_) [63].

Sublethal concentrations of insecticides can affect the development and growth of pests, resulting in a decrease in population density; therefore, evaluating not only the insecticidal activity but also the sublethal effects is important.

This study is expected to be helpful in developing a pest management program for effective *A. gossypii* control via the sublethal effects of spirotetramat, cyantraniliprole, and pymetrozine.

## 5. Conclusions

We studied the sublethal effects of three insecticides (spirotetramat, cyantraniliprole, and pymetrozine) on *A. gossypii*. The results for the life table parameters indicated that *A. gossypii* can be controlled via population density management using sublethal concentrations of these insecticides.

## Figures and Tables

**Figure 1 insects-15-00247-f001:**
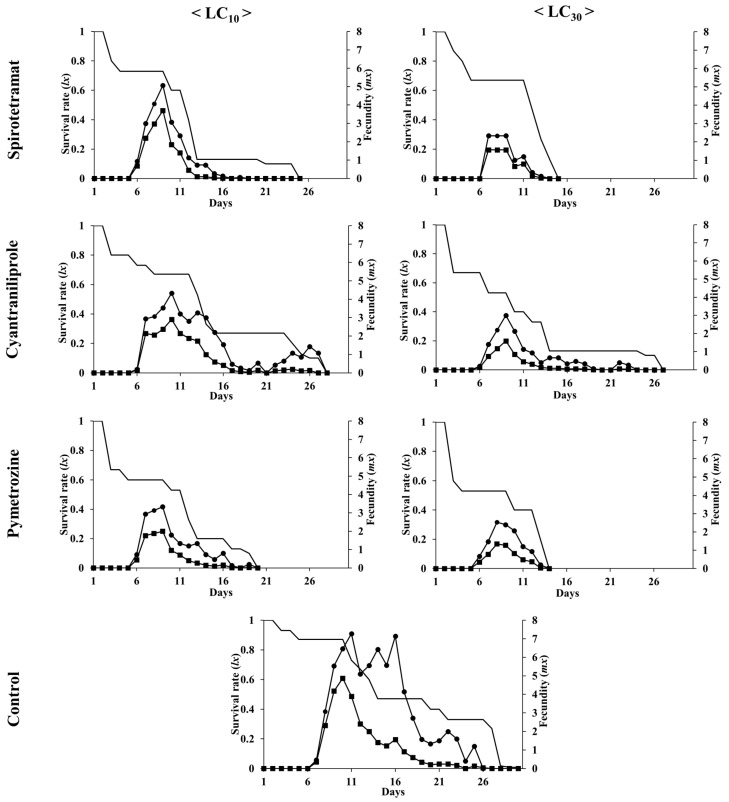
Age-specific survival rate (*l_x_,*
**━**), fecundity (*m_x_*, ●), and net maternity (*l_x_m_x_*, ■) of *A. gossypii* exposed to sublethal concentrations (LC_10_ and LC_30_) of three insecticides and the control group.

**Table 1 insects-15-00247-t001:** Toxicity of spirotetramat, cyantraniliprole, and pymetrozine to *A. gossypii* nymphs and adults.

Insecticides	Stage	*n* ^a^	DAT ^b^	LC_10_	RT ^d^	LC_30_	RT	LC_50_	RT	LC_70_	RT	LC_90_	RT	Slope ± SE	*df*	*p*-Value
				(95% CL ^c^)		(95% CL)		(95% CL)		(95% CL)		(95% CL)				
Spirotetramat	Nymph	677	4	0.07 (0.05–0.10)	8.0	0.23 (0.18–0.29)	8.2	0.52 (0.43–0.61)	8.9	1.14 (0.96–1.37)	8.9	3.58 (2.80–4.84)	9.5	1.53 ± 0.10	6	<0.0001
	Adult	500	4	0.56 (0.38–0.77)		1.89 (1.47–2.33)		4.36 (3.62–5.21)		10.09 (8.37–12.45)		33.89 (25.76–47.68)		1.44 ± 0.10	5	<0.0001
Cyantraniliprole	Nymph	599	3	0.04 (0.03–0.05)	2.5	0.15 (0.12–0.18)	3.3	0.38 (0.32–0.46)	3.9	1.00 (0.81–1.25)	4.5	3.99 (2.95–5.75)	5.4	1.26 ± 0.07	7	<0.0001
	Adult	478	3	0.10 (0.06–0.15)		0.50 (0.37–0.65)		1.50 (1.20–1.88)		4.47 (3.50–5.92)		21.61 (1.94–34.38)		1.11 ± 0.08	5	<0.0001
Pymetrozine	Nymph	571	3	0.34 (0.21–0.51)	1.2	1.76 (1.30–2.26)	1.4	5.46 (4.42–6.68)	1.7	16.95 (13.68–21.56)	1.9	86.95 (62.22–131.65)	2.3	1.07 ± 0.07	7	<0.0001
	Adult	501	3	0.41 (0.22–0.66)		2.55 (1.79–3.48)		9.08 (6.93–11.78)		32.34 (24.48–44.45)		202.27 (132.31–345.99)		0.95 ± 0.07	5	<0.0001

^a^ *n*, Number of *Aphis gossypii*. ^b^ DAT, Day after treatment. ^c^ 95% Confidence limits. ^d^ RT, Relative toxicity to nymph and adult at each insecticide.

**Table 2 insects-15-00247-t002:** Effects of sublethal and lethal concentrations of the three insecticides on *A. gossypii* nymphs.

Insecticides	Treatment	P				F_1_				F_2_
		Developmental Period (Day)	Adult Survival Rate (%)	Adult Longevity (Day)	Fecundity (%)	Deformity Rate (%)	Developmental Period (Day)	Adult Survival Rate (%)	Fecundity (%)	Deformity Rate (%)
Spirotetramat	Control	5.35 ± 0.16 b	96.30 ± 0.55 a	16.01 ± 0.91 a	100.0 ± 0.00 a	0.0 ± 0.00 c	5.06 ± 0.17 c	83.04 ± 2.51 a	100.0 ± 0.00 a	0.0 ± 0.00
	LC_10_	5.52 ± 0.07 b	91.60 ± 0.84 a	10.53 ± 0.59 b	43.24 ± 4.43 b	0.0 ± 0.00 c	5.40 ± 0.17 c	82.56 ± 2.40 a	63.29 ± 10.95 b	
	LC_30_	5.40 ± 0.03 b	78.07 ± 1.39 b	9.25 ± 1.05 b	24.72 ± 1.86 c	19.33 ± 1.93 b	6.22 ± 0.12 b	56.57 ± 3.64 b	42.40 ± 5.58 c	
	LC_50_	6.03 ± 0.09 a	54.93 ± 2.78 c	10.17 ± 0.88 b	9.72 ± 1.32 d	24.13 ± 1.70 b	6.99 ± 0.14 ab	24.24 ± 4.10 c	18.77 ± 4.12 d	
	LC_70_	6.10 ± 0.08 a	22.73 ± 2.94 d	9.49 ± 0.45 b	9.32 ± 0.64 d	59.55 ± 7.23 a	6.65 ± 0.23a	23.47 ± 2.44 c	5.63 ± 2.82 d	
Cyantraniliprole	Control	5.17 ± 0.12 a	93.20 ± 2.11 a	15.26 ± 0.57 a	100.0 ± 0.00 a	0.0 ± 0.00 b	5.09 ± 0.49 a	85.79 ± 3.35 a	100.0 ± 0.00 a	
	LC_10_	5.72 ± 0.07 a	87.00 ± 3.06 a	12.93 ± 0.93 b	88.19 ± 3.53 b	1.07 ± 0.27 b	5.15 ± 0.08 a	73.15 ± 3.20 a	91.67 ± 8.80 a	
	LC_30_	5.65 ± 0.26 a	69.40 ± 3.66 b	9.56 ± 0.13 c	41.64 ± 0.63 c	4.54 ± 0.20 b	5.14 ± 0.14 a	51.01 ± 1.24 b	51.83 ± 5.83 a	
	LC_50_	5.48 ± 0.02 a	67.77 ± 3.39 b	9.45 ± 0.49 c	18.45 ± 0.75 d	15.73 ± 2.17 a	5.84 ± 0.20 a	42.48 ± 7.59 b	36.65 ± 2.96 b	
	LC_70_	5.60 ± 0.25 a	38.87 ± 1.30 c	8.41 ± 0.11 c	15.58 ± 1.00 d	18.15 ± 4.40 a	5.91 ± 0.09 a	21.51 ± 2.43 c	21.48 ± 5.10 b	
Pymetrozine	Control	5.38 ± 0.26 ab	92.33 ± 2.10 a	16.41 ± 1.62 a	100.0 ± 0.00 a	0.0 ± 0.00 a	5.09 ± 0.29 a	84.63 ± 3.00 a	100.0 ± 0.00 a	
	LC_10_	5.23 ± 0.03 b	84.93 ± 3.87 a	9.19 ± 1.15 b	45.63 ± 4.55 b	0.0 ± 0.00 a	5.30 ± 0.07 a	71.99 ± 2.00 b	77.55 ± 3.54 b	
	LC_30_	5.43 ± 0.09 ab	70.33 ± 3.25 b	7.21 ± 0.61 b	42.19 ± 1.63 b	0.0 ± 0.00 a	5.46 ± 0.15 a	62.23 ± 1.81 b	51.90 ± 4.85 c	
	LC_50_	5.80 ± 0.18 ab	56.40 ± 3.16 c	6.44 ± 0.16 b	18.17 ± 0.78 c	0.0 ± 0.00 a	5.65 ± 0.23 a	34.78 ± 4.82 c	37.25 ± 8.90 c	
	LC_70_	5.83 ± 0.19 a	28.80 ± 4.77 d	6.16 ± 0.02 b	14.43 ± 1.36 c	0.0 ± 0.00 a	5.73 ± 0.29 a	35.12 ± 4.35 c	17.30 ± 2.10 d	

Mean values in the same column for each insecticide followed by the same letters are not significantly different at *p* < 0.05 (Duncan’s test).

**Table 3 insects-15-00247-t003:** Effects of sublethal and lethal concentrations of the three insecticides on *A*. *gossypii* adults.

Insecticides	Treatment	P			F_1_				F_2_
		Adult Survival Rate (%)	Adult Longevity (Day)	Fecundity (%)	Deformity Rate (%)	Developmental Period (Day)	Adult Survival Rate (%)	Fecundity (%)	Deformity Rate (%)
Spirotetramat	Control	92.77 ± 1.63 a	13.85 ± 0.88 a	100.0 ± 0.0 a	0.0 ± 0.00 d	4.73 ± 0.29 d	95.97 ± 1.16 a	100.0 ± 0.0 a	0.0 ± 0.00
	LC_10_	70.30 ± 2.46 b	8.03 ± 0.27 b	68.88 ± 6.33 b	0.0 ± 0.00 d	5.43 ± 0.15 c	92.03 ± 0.52 b	47.91 ± 3.14 b	
	LC_30_	40.67 ± 2.85 c	6.64 ± 0.39 bc	38.28 ± 2.30 c	25.46 ± 5.22 c	6.25 ± 0.10 b	66.27 ± 0.90 c	30.34 ± 2.58 c	
	LC_50_	23.40 ± 0.40 d	6.36 ± 0.87 bc	31.77 ± 7.09 cd	51.38 ± 5.63 b	6.85 ± 0.23 ab	41.43 ± 1.45 d	22.05 ± 2.07 d	
	LC_70_	18.07 ± 3.67 d	5.13 ± 0.33 c	21.69 ± 0.35 d	63.79 ± 2.63 a	6.65 ± 0.23 a	13.53 ± 1.12 e	20.47 ± 2.68 d	
Cyantraniliprole	Control	90.17 ± 0.17 a	12.48 ± 0.75 a	100.0 ± 0.0 a	0.0 ± 0.00 c	4.62 ± 0.26 b	92.37 ± 0.78 a	100.0 ± 0.0 a	
	LC_10_	78.97 ± 1.71 b	11.75 ± 0.41 b	63.32 ± 8.17 b	0.0 ± 0.00 c	5.29 ± 0.07 a	90.90 ± 0.95 a	59.80 ± 7.75 b	
	LC_30_	52.73 ± 1.87 b	9.49 ± 0.84 bc	40.54 ± 3.40 c	19.42 ± 2.15 b	5.46 ± 0.15 a	73.07 ± 1.97 b	33.20 ± 5.41 c	
	LC_50_	52.17 ± 3.00 c	7.50 ± 0.61 c	34.17 ± 0.89 c	14.83 ± 1.14 b	5.65 ± 0.23 a	61.07 ± 1.94 c	31.50 ± 3.71 c	
	LC_70_	37.07 ± 4.30 c	7.38 ± 0.36 c	38.95 ± 7.14 c	34.85 ± 3.24 a	5.73 ± 0.29 a	18.27 ± 1.75 d	30.01 ± 2.12 c	
Pymetrozine	Control	90.20 ± 2.52 a	13.08 ± 2.20 a	100.0 ± 0.0 a	0.0 ± 0.00 a	4.68 ± 0.32 b	92.57 ± 2.09 a	100.0 ± 0.0 a	
	LC_10_	67.40 ± 3.85 b	8.45 ± 1.06 a	57.35 ± 6.46 b	0.0 ± 0.00 a	5.15 ± 0.08 b	87.83 ± 1.16 a	40.23 ± 3.99 b	
	LC_30_	66.07 ± 2.72 c	6.59 ± 0.34 ab	40.08 ± 11.66 bc	0.0 ± 0.00 a	5.14 ± 0.14 b	78.53 ± 1.79 b	21.14 ± 1.75 c	
	LC_50_	37.17 ± 3.42 c	6.16 ± 0.50 b	30.19 ± 0.55 c	0.0 ± 0.00 a	5.84 ± 0.20 a	59.80 ± 1.80 c	27.03 ± 3.22 c	
	LC_70_	36.23 ± 5.06 d	5.77 ± 0.25 b	36.56 ± 4.02 c	0.0 ± 0.00 a	5.91 ± 0.09 a	22.40 ± 1.47 d	23.35 ± 2.83 c	

Mean values in the same column for each insecticide followed by the same letters are not significantly different at *p* < 0.05 (Duncan’s test).

**Table 4 insects-15-00247-t004:** Fertility life table parameters for *A. gossypii* nymphs at sublethal concentrations of the three insecticides.

Insecticides	Treatment	*R* _0_	*T*	*r_m_*	*λ*	*DT*
Spirotetramat	LC_10_	18.01 ± 2.53 bc	8.75 ± 0.56 b	0.32 ± 0.02 ab	1.37 ± 0.01 ab	2.32 ± 0.21 a
	LC_30_	10.15 ± 1.37 d	8.62 ± 0.17 b	0.25 ± 0.02 c	1.29 ± 0.03 c	2.84 ± 0.24 a
	*p*-value	0.0073	0.6109	0.0171	0.0145	0.0366
Cyantraniliprole	LC_10_	24.20 ± 4.44 b	10.41 ± 0.32 a	0.26 ± 0.04 bc	1.30 ± 0.01 c	3.26 ± 1.06 a
	LC_30_	11.12 ± 1.50 cd	9.14 ± 0.24 b	0.25 ± 0.02 c	1.29 ± 0.02 c	2.90 ± 0.11 a
	*p*-value	0.0405	0.0818	0.9947	0.709	0.4551
Pymetrozine	LC_10_	14.32 ± 1.85 cd	8.74 ± 0.34 b	0.30 ± 0.01 abc	1.35 ± 0.01 abc	2.33 ± 0.05 a
	LC_30_	11.82 ± 0.64 cd	8.64 ± 0.20 b	0.29 ± 0.04 bc	1.33 ± 0.04 bc	2.56 ± 0.12 a
	*p*-value	0.2192	0.9601	0.1729	0.2055	0.1669
Control	-	38.58 ± 3.72 a	10.66 ± 0.58 a	0.35 ± 0.01 a	1.42 ± 0.02 a	2.01 ± 0.07 a

*R*_0_ (NRR): Net reproductive rate; *r_m_*: Intrinsic rate of population increase; *λ*: Finite rate of population increase; *T*: Generation time; *DT*: Doubling time. Mean values in a same column of each insecticide by the same letters are not significantly different at *p* < 0.05 (Duncan’s test).

## Data Availability

Data is contained within the article.

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
