# Peer review of "Sublethal Effects of Spirotetramat, Cyantraniliprole, and Pymetrozine on Aphis gossypii (Hemiptera: Aphididae)"

_insects, 2024, doi:10.3390/insects15040247_

Round 1

Reviewer 1 Report

Comments and Suggestions for Authors

In the present study, the author investigated the sublethal effects of spirotetramat, cyantraniliprole, and pymetrozine on Aphis gossypii. The susceptibility of nymphal stage to the three insecticides  was up to 8.9-fold than the adult stage. Nonviable nymphs occurred in the F1 generation when both nymph and adults were treated with spirotetramat and cyantraniliprole. The sublethal concentrations (over the LC30) of spirotetramat, cyantraniliprole and pymetrozine are useful for the density management of A. gossypii. The results provided reference for the chemical control of A. gossypii in the field. I think this manuscript is suitable for publication in insects after revisions.

Here are my several concerns with the manuscript:

1. In the Introduction, the paragraphs about the introduction of three insecticides should be cut down and merged, meanwhile add paragraphs about the research progress on chemical control of A. gossypii, and the sublethal effects.

2. The concentration gradients of each insecticides used in the bioassay should be added in 2.3 Insecticidal toxicity experiment.

3. Why the fecundity in F1 generation of the control was 0 in Table 3?

Comments on the Quality of English Language

Although the English language of the manuscript has been edited by native English speaking editors at AJE, the details  still need minor editing.

Author Response

R1

Comments and Suggestions for Authors

In the present study, the author investigated the sublethal effects of spirotetramat, cyantraniliprole, and pymetrozine on Aphis gossypii. The susceptibility of nymphal stage to the three insecticides  was up to 8.9-fold than the adult stage. Nonviable nymphs occurred in the F1 generation when both nymph and adults were treated with spirotetramat and cyantraniliprole. The sublethal concentrations (over the LC30) of spirotetramat, cyantraniliprole and pymetrozine are useful for the density management of A. gossypii. The results provided reference for the chemical control of A. gossypii in the field. I think this manuscript is suitable for publication in insects after revisions.

Here are my several concerns with the manuscript:

  1. In the Introduction, the paragraphs about the introduction of three insecticides should be cut down and merged, meanwhile add paragraphs about the research progress on chemical control of  gossypii, and the sublethal effects.

==> inserted more description of sublethal effects

  1. The concentration gradients of each insecticides used in the bioassay should be added in “2.3 Insecticidal toxicity experiment”.

==> inserted df and p-values into table and detailed concentrations in the M&M

  1. Why the fecundity in F1 generation of the control was 0 in Table 3?

==> Changed 100

Comments on the Quality of English Language

Although the English language of the manuscript has been edited by native English speaking editors at AJE, the details  still need minor editing.

==> re-edited

Reviewer 2 Report

Comments and Suggestions for Authors

The manuscript entitled “Sublethal effects of spirotetramat, cyantraniliprole, and pymetrozine on Aphis gossypii (Hemiptera: Aphididae)” by Kim et al. addresses the sublethal effects of several doses (LC10, LC30, LC50, and LC70) on the basic biological features of Aphis gossypii, covering parental, F1, and F2 generations and indicating transgenerational effects. Additionally, the authors explore the effects of two sublethal doses (LC10, LC30) of three pesticides on life table parameters. While the manuscript provides useful information for pest management, I have several major concerns that I have itemized below.

  1. The authors should provide details on the methods used to determine lethal concentrations. Although they mention using probit analysis, they have not provided the goodness-of-fit for the model, including values such as P and df. The provided chi-square values are unacceptably high, indicating a poor fit. The goodness-of-fit probability (P) should be provided; a value higher than 0.05 would indicate that the model fits the data.
  2. The authors should also provide the concentrations tested in units such as ml/l, and include information on the recommended field rate for these three pesticides against the pest.
  3. The authors need to clearly indicate the software used for calculating standard errors for life table parameters (e.g., r, D, T, Ro). Additionally, the method used for standard error calculation (e.g., bootstrap, jackknife) is omitted. The authors used Tukey’s method for multiple comparisons, which is not widely accepted. Instead, they should have employed paired bootstrap for all calculations and comparisons.
  4. The minimum number of replications for life table analysis is typically 15, but this study was conducted with only 10 replications for each sublethal dose.
  5. Authors should cite more relevant earlier studies on sublethal effects of pesticides on A. gossypii or other aphid species, pests having similar mouth parts (piercing-sucking) first then insects belonging to other insect Orders. I suggest using web of science (Clavirate) or Scopus to check more related papers by appropriate keywords.

Miscellaneous Suggestions:

• L73. Please indicate the tested dose series. Also, let readers know the field rate of these insecticides.

• L106. The authors should clarify which software was used for life table analysis and how they generated the standard errors of the means—was it through jackknife or bootstrap methods? If the latter, please inform the readers about the number of replications (e.g., 100,000) and specify the software used for the bootstrap analysis. Was the Two-sex chart utilized for this purpose?

• L109. Let readers know by which model (probit, logit) and which transformation method (log base10, natural log) used to fit the model for the calculation of lethal concentrations. Please also provide goodness of fit Chi-Square, df and P values as proof of the fitness of your probit model.

• L112. The two-sex chart does not perform Tukey studentized range test for comparisons but uses paired bootstrap test.

• Table 1. Please check Chi-Square and provide df and P here.

• L138. Please provide F, df, and P values for all comparisons.

• L171. Please provide F, df, P values for all comparisons. Because the authors did not indicate which software used for life table parameters comparisons, two-sex chart is widely accepted for life table analysis. I suggest the authors use this software.

Author Response

We would also like to take this opportunity to express our thanks to the reviewer for the positive feedback and helpful comments for correction or modification.

We are corrected several words and inserted more detail.

It was supported for the re-edition of English at American Journal Experts.

R2

Comments and Suggestions for Authors

The manuscript entitled “Sublethal effects of spirotetramat, cyantraniliprole, and pymetrozine on Aphis gossypii (Hemiptera: Aphididae)” by Kim et al. addresses the sublethal effects of several doses (LC10, LC30, LC50, and LC70) on the basic biological features of Aphis gossypii, covering parental, F1, and F2 generations and indicating transgenerational effects. Additionally, the authors explore the effects of two sublethal doses (LC10, LC30) of three pesticides on life table parameters. While the manuscript provides useful information for pest management, I have several major concerns that I have itemized below.

  1. The authors should provide details on the methods used to determine lethal concentrations. Although they mention using probit analysis, they have not provided the goodness-of-fit for the model, including values such as P and df. The provided chi-square values are unacceptably high, indicating a poor fit. The goodness-of-fit probability (P) should be provided; a value higher than 0.05 would indicate that the model fits the data.

  • Inserted df and p-values

  1. The authors should also provide the concentrations tested in units such as ml/l, and include information on the recommended field rate for these three pesticides against the pest.

  • Inserted concentrations of insecticides

  1. The authors need to clearly indicate the software used for calculating standard errors for life table parameters (e.g., r, D, T, Ro). Additionally, the method used for standard error calculation (e.g., bootstrap, jackknife) is omitted. The authors used Tukey’s method for multiple comparisons, which is not widely accepted. Instead, they should have employed paired bootstrap for all calculations and comparisons.
  • There was the wrong explanation due to my misunderstanding. The explanation of ‘two-sex life table theory’ was deleted and the results calculated by citing Wu et al., (Agronomy 2022, 12, 1334).

  1. The minimum number of replications for life table analysis is typically 15, but this study was conducted with only 10 replications for each sublethal dose.

è We wish we had done more, but thought it was enough.

  1. Authors should cite more relevant earlier studies on sublethal effects of pesticides on A. gossypii or other aphid species, pests having similar mouth parts (piercing-sucking) first then insects belonging to other insect Orders. I suggest using web of science (Clavirate) or Scopus to check more related papers by appropriate keywords.
  • Inserted more references in introduction.

Miscellaneous Suggestions:

  • L73. Please indicate the tested dose series. Also, let readers know the field rate of these insecticides.
  • Inserted the concentrations and this experiment was conducted using single chemical agent (active ingredient).
  • L106. The authors should clarify which software was used for life table analysis and how they generated the standard errors of the means—was it through jackknife or bootstrap methods? If the latter, please inform the readers about the number of replications (e.g., 100,000) and specify the software used for the bootstrap analysis. Was the Two-sex chart utilized for this purpose?
  • There was the wrong explanation due to my misunderstanding. The explanation of ‘two-sex life table theory’ was deleted and the results calculated by citing Wu et al., (Agronomy 2022, 12, 1334).
  • L109. Let readers know by which model (probit, logit) and which transformation method (log base10, natural log) used to fit the model for the calculation of lethal concentrations. Please also provide goodness of fit Chi-Square, df and P values as proof of the fitness of your probit model.
  • inserted
  • L112. The two-sex chart does not perform Tukey studentized range test for comparisons but uses paired bootstrap test.
  • There was the wrong explanation due to my misunderstanding. The explanation of ‘two-sex life table theory’ was deleted
  • Table 1. Please check Chi-Square and provide df and P here.
  • Changed
  • L138. Please provide F, df, and P values for all comparisons.

è inserted inmanuscript

  • L171. Please provide F, df, P values for all comparisons. Because the authors did not indicate which software used for life table parameters comparisons, two-sex chart is widely accepted for life table analysis. I suggest the authors use this software.
  • Deleted and edited

Reviewer 3 Report

Comments and Suggestions for Authors

The laboratory study reports the effects of 3 insecticides exposure on the cotton aphid  A. gossyppi. The data presented could be useful for studies, due to the importance of the study of toxic effect of the insecticide on one of the most important pest. Nevertheless, the manuscript presents some incongruences that must be corrected for a possible publication. i) The methodology cannot be reproduced by other scientists, because it is incomplete; (ii) The introduction and discussion has a lack of information about the effects of insecticides tested on pest or predator, and the probable effects on A. gossypi physiology was poorly explored.

-Keywords should be in alphabetic order. Also, keywords serve to widen the opportunity to be retrieved from a database. To put words that already are into title and abstracts makes KW not useful. Please choose terms that are neither in the title nor in abstract.

-Small N of insect tested

-Please give some examples here about toxic or lethal effects to other pest or predators (eg coccinellids) bees etc.

The authors should add concentrations used in order to find LC results.

Author Response

We would also like to take this opportunity to express our thanks to the reviewer for the positive feedback and helpful comments for correction or modification.

We are corrected several words and inserted more detail.

It was supported for the re-edition of English at American Journal Experts.

R3

Comments and Suggestions for Authors

The laboratory study reports the effects of 3 insecticides exposure on the cotton aphid  A. gossyppi. The data presented could be useful for studies, due to the importance of the study of toxic effect of the insecticide on one of the most important pest. Nevertheless, the manuscript presents some incongruences that must be corrected for a possible publication. i) The methodology cannot be reproduced by other scientists, because it is incomplete; (ii) The introduction and discussion has a lack of information about the effects of insecticides tested on pest or predator, and the probable effects on A. gossypi physiology was poorly explored.

-Keywords should be in alphabetic order. Also, keywords serve to widen the opportunity to be retrieved from a database. To put words that already are into title and abstracts makes KW not useful. Please choose terms that are neither in the title nor in abstract.

 è Changed

-Small N of insect tested

è I think it’s enough compared to other papers (30 insects 10 replicates).

-Please give some examples here about toxic or lethal effects to other pest or predators (eg coccinellids) bees etc.

è In the introduction and discussion, some negative effects due to sublethal effects on pests and natural enemies are described.

The authors should add concentrations used in order to find LC results.

  • Inserted

Thank you very much.

Round 2

Reviewer 2 Report

Comments and Suggestions for Authors

After checking improvements and responses provided by the authors in revised version, I can clearly say the authors did their best, however, I would like authors to know Duncan’s multiple range test is not an appopriate test. Duncan’s multiple range test developed for agronomic studies and is a very sensitive test. And I still believe the best and widely aceepted, life table analysis can be done by two sex chart software.